# Identifying and Evaluating Young Children with Developmental Central Hypotonia: An Overview of Systematic Reviews and Tools

**DOI:** 10.3390/healthcare12040493

**Published:** 2024-02-18

**Authors:** Álvaro Hidalgo Robles, Ginny S. Paleg, Roslyn W. Livingstone

**Affiliations:** 1Facultad de Educación, Universidad Internacional de La Rioja, 26006 Logroño, Spain; 2Physical Therapist, Montgomery County Infants and Toddlers Program, Rockville, MD 20825, USA; ginny@paleg.com; 3Occupational Science and Occupational Therapy, Faculty of Medicine, University of British Columbia, Vancouver, BC V6T 2B5, Canada; roslyn.livingstone@ubc.ca

**Keywords:** measurement, evaluation, classification, diagnosis, validity, reliability, down syndrome, low tone, hypotonic cerebral palsy

## Abstract

Children with developmental central hypotonia have reduced muscle tone secondary to non-progressive damage to the brain or brainstem. Children may have transient delays, mild or global functional impairments, and the lack of a clear understanding of this diagnosis makes evaluating appropriate interventions challenging. This overview aimed to systematically describe the best available evidence for tools to identify and evaluate children with developmental central hypotonia aged 2 months to 6 years. A systematic review of systematic reviews or syntheses was conducted with electronic searches in PubMed, Medline, CINAHL, Scopus, Cochrane Database of Systematic Reviews, Google Scholar, and PEDro and supplemented with hand-searching. Methodological quality and risk-of-bias were evaluated, and included reviews and tools were compared and contrasted. Three systematic reviews, an evidence-based clinical assessment algorithm, three measurement protocols, and two additional measurement tools were identified. For children aged 2 months to 2 years, the Hammersmith Infant Neurological Examination has the strongest measurement properties and contains a subset of items that may be useful for quantifying the severity of hypotonia. For children aged 2–6 years, a clinical algorithm and individual tools provide guidance. Further research is required to develop and validate all evaluative tools for children with developmental central hypotonia.

## 1. Introduction

Muscle tone or tonus is typically taken to mean tonic anti-gravity muscle activity or resistance to passive movement [1]. The term is not well defined, and evaluation is typically subjective [2]. Decreased resistance to movement observed in individuals with low muscle tone or hypotonia is not due to an absent or decreased stretch response [1] but may arise from a complex interaction between neural circuits in the muscle spindle, the spinal cord, and the brain [3]. Hypotonia can best be described as difficulties in sustaining postural control and moving against gravity (active muscle tone) as well as decreased resistance to passive movement [4].

Decreased resistance to passive movement in arms and legs is described as phasic or resting tone, while postural or active tone is observed in the axial muscles of the neck, back, and trunk. Several clinical maneuvers including “pull to sit” (also referred to as ‘head lag’), “scarf sign” (also referred to as ‘shoulder adduction’), “shoulder suspension” (also referred to as ‘vertical suspension’ or ‘slip through hands’), and “ventral suspension” (also referred to as ‘rag doll posture’ or ‘horizontal/prone suspension’) are commonly used to identify hypotonia in infants and young children [5].

Hypotonia is commonly divided into two main types. Central hypotonia is secondary to supra-spinal/supra-segmental damage in the brain, brainstem, or cervical spinal junction. It includes infants with systemic disease, encephalopathies, genetic syndromes, and brain abnormalities as well as delayed development without overt brain pathology. Peripheral or motor-unit hypotonia includes damage to the anterior horn cells (e.g., spinal muscular atrophy), peripheral nerves, neuromuscular junction, or muscle [5]. Central hypotonia occurs in approximately 60–80% of cases and includes children with transient hypotonia due to prematurity, drug exposure, illness or infections, as well as developmental disorders such as Down or Prader-Willi syndromes. There are also some progressive or degenerative conditions that combine central and peripheral origins [6].

The vast majority of articles discussing hypotonia focus on distinguishing between central and peripheral hypotonia in neonates or very young children. In contrast, little attention has been paid to the causes or management of developmental central hypotonia [7]. This manuscript will focus on children two months and older with persistent hypotonia who may be referred to early intervention occupational therapy (OT) or physical therapy (PT). In infants with central causes of hypotonia, weakness is very unusual except in acute illness and/or in the neonatal period, and profound weakness usually indicates a peripheral cause [8]. Older infants and children with developmental central hypotonia may have mild to moderate muscle weakness, decreased or increased (but not absent) deep tendon reflexes [9], and, in addition, functional abilities and developmental trajectory may vary widely. For example, Palisano and colleagues demonstrated two distinct motor trajectories (mild and moderate/severe) in young children with Down syndrome [10].

Infants with CP due to hypoxic ischaemic encephalopathies initially present with hypotonia. Most develop hypertonia in the limbs over time, although the trunk and neck may remain hypotonic [4]. Depending on the severity of the functional presentation, some children with hypotonia due to genetic syndromes or other developmental disabilities meet the clinical criteria for a CP diagnosis [11], while some have intellectual and sensory impairments but not a primary motor disability, and others ‘catch up’ with their peers by school-age [5]. Developmental trajectories may also be divided into those who have a global impairment, those with mild impairment (coordination, language, or learning disabilities), and those with transient impairment [12]. Internationally there is inconsistency in the inclusion of children with hypotonia (with or without additional genetic diagnoses) in CP registries, with less than half including children with the sole motor type of hypotonia [13]. This has led to a paucity of information about developmental central hypotonia.

Currently, there is no agreement on how hypotonia should be measured in children [14,15,16]. This issue has been raised in the literature through surveys [17,18,19], expert consensus studies [20], proposals [14,21,22,23], and a clinical algorithm has been developed [24]. However, as yet, no hypotonia measurement tools that are reliable or valid for use in clinical practice have been recognized or widely implemented.

A valid and reliable means of quantifying hypotonia in this population is needed, primarily for evaluative purposes, since PT and OT interventions may be anticipated to enhance activity and participation but not necessarily change underlying muscle tone. It is essential to measure both postural (active) and phasic (passive or resting) muscle tone. Quantifying the severity of presentation, may help to identify the potential functional trajectory, and assist in determining the most appropriate or beneficial interventions.

The purpose of this overview of systematic reviews or syntheses is to systematically describe the best available evidence regarding characteristics, tools, and methods for identification and evaluation of hypotonia in young children with non-degenerative developmental central hypotonia aged between 2 months and 6 years.

Specific sub-questions are as follows:What characteristics, tools or measures for children with developmental central hypotonia have been examined in systematic reviews or evidence syntheses?What is the methodological quality and/or risk-of-bias of the included studies?How do reviews or syntheses compare or contrast?What are the measurement properties of tools to evaluate hypotonia?What recommendations can be made regarding evaluation and quantification of hypotonia for use in clinical practice and the need for further research?

## 2. Materials and Methods

The review protocol was registered in Open Science Framework on 17th November 2023 (https://osf.io/76k4c/?view_only=80d9996daa644519a666adfa2225972d, accessed on 13 February 2024). The PRISMA 2020 statement [25] guided this review, along with published guidance for conducting umbrella reviews [26] or overviews of intervention effectiveness [27,28].

### 2.1. Search Strategy

Database searches were conducted during November 2023 and included PubMed, Medline and CINAHL (via EBSCO), Cochrane Library (via OVID), SCOPUS, PEDro, and Google Scholar. No restrictions were placed on language in any database. PubMed, Medline, and CINAHL searches were limited to the years 2000 to November 2023, and the PubMed search was limited to systematic reviews. Reporting standards and methodologies for conducting systematic reviews have changed over the last two decades and this date was chosen to ensure that reviews would meet contemporary standards and be suitable to combine in an overview. Searches in other databases were not restricted in order to ensure identification of any potentially relevant tools regardless of publication type or date. Search terms included ‘hypotonia’ or ‘low muscle tone’ or ‘muscle, hypotonia’ or ‘muscle tone’ AND ‘assess*’ or ‘evaluat*’ or ‘measure*’ or ‘test’* or ‘examin*’. See Appendix B for the search strategy.

In this manuscript, we use the term developmental central hypotonia to include only conditions where hypotonia is secondary to a non-degenerative brain impairment. This includes a variety of genetic developmental conditions, hypotonic cerebral palsy (CP), developmental delay and so-called benign congenital hypotonia, also known as hypotonia with a favorable outcome [7]. We exclude damage to the spinal cord and progressive/degenerative conditions since diagnosis, prognosis, and interventions differ.

Reviews or syntheses were read full-text if the abstract or title indicated that measurement tools or methods for the assessment and evaluation of muscle tone (including hypotonia) in children were examined. Systematic reviews of interventions for children with developmental central hypotonia were read full-text to determine whether hypotonia measurement tools were included. Articles describing potential tools or measures that may not have been evaluated in systematic reviews or syntheses were also read full-text. Citation searches of included full-text articles were completed, and secondary Google Scholar and PubMed searches were conducted for specific tools identified in the primary database search.

Systematic reviews, syntheses, and individual articles were excluded in the following cases: (1) They reviewed measures exclusively for children with hypertonia; (2) Data on tools related to the assessment of children with developmental central hypotonia could not be extracted; (3) Measures or tools required specialized equipment (e.g., electromyography or myotonometer) or a laboratory setting (e.g., 3D motion analysis) and could not be implemented in a typical home or community therapy setting; (4) They focused on methods only to distinguish between central and peripheral hypotonia or specifically for peripheral hypotonia; (5) Tools only evaluated passive tone or muscle stretch.

### 2.2. Search and Screening Process

Database searches were conducted by one author (RWL) with strategy reviewed by the other two authors. Hand-searching of reference lists and intervention systematic reviews and secondary searches for specific tools were conducted by two authors (RWL and GSP). Titles and abstracts were exported to the online tool Rayyan [29] (http://rayyan.qcri.org, accessed on 17 November 2023) where they were screened by at least two of three authors independently. Any disagreements on articles to be read full-text were resolved through discussion.

Full-text articles were loaded into Rayyan for independent screening. All three authors reviewed full-text articles independently with reasons for inclusion/exclusion documented. Inclusion and exclusion of reviews, syntheses, and individual measures were agreed through discussion.

### 2.3. Data Extraction and Synthesis

The umbrella review data extraction form [30] and quality appraisal checklist for systematic reviews and research syntheses [31] from the Joanna Briggs Institute and Risk-of-Bias in Systematic reviews (ROBIS) [32] ratings were completed for each review. The Appraisal of Guidelines for Research and Evaluation version two (AGREE II) [33] was used for quality appraisal of clinical guidelines or syntheses.

The McMaster critical review form for quantitative studies [34] was used to extract data from any single studies not examined in systematic reviews, and study conduct was appraised using the Mixed Methods Assessment Tool (MMAT 2018) [35] where appropriate. Although measures or characteristics for the evaluation of hypotonia in children are not sufficiently developed to implement the COnsensus-based Standards for the selection of health status Measurement INstruments (COSMIN) guideline for systematic reviews of measurement tools [36], we sought to update measurement property evaluation initiated in included reviews.

For additional studies (not evaluated in reviews) that reported measurement properties of relevant tools, the McMaster Outcome Measures Rating Form (OMRF) [37] was used for data extraction, and risk-of-bias was evaluated using COSMIN checklists. The recently developed risk-of-bias checklist for studies of reliability or measurement error in clinician-reported measures [38] was used to evaluate evidence related to inter- or intra-rater reliability. The risk-of-bias checklist for patient-reported outcome measures [39] continues to be recommended as appropriate for assessments of validity in clinician-reported measures and was used to evaluate studies reporting convergent or predictive (criterion) validity or hypotheses testing (construct validity). COSMIN group’s modified GRADE approach to evaluating the quality of evidence [40] was used to summarize evidence for the reliability and validity of included tools where appropriate. For tools or scales where formal studies of measurement properties had not been published, OMRF terminology was used to describe evidence. Data extraction and evaluations of risk-of-bias and quality were completed independently by all authors and then combined and agreed through discussion.

## 3. Results

The primary electronic search identified 900 articles. After the removal of duplicates, and title and abstract screening, 35 were agreed to be reviewed full-text. Twelve articles [14,15,18,20,21,22,23,24,41,42,43,44] reporting seven distinct studies met the inclusion criteria. In addition, two other hypotonia measurement scales were identified through citation and hand-searches. One was published in a textbook [45] and the other in two different conference publications [46,47].

Twenty-three articles were excluded following primary database full-text review: eleven articles did not describe hypotonia measurement [48,49,50,51,52,53,54,55,56,57,58]; five articles described some aspects of hypotonia measurement but had already been fully evaluated in an included review or synthesis [1,5,6,17,19]; five articles described the assessment of passive muscle tone only [59,60,61,62,63]; one article described hypertonia assessment only [64]; and, in addition, a scoping review of muscle tone evaluations [65] included only 2/84 studies evaluating hypotonia. Citation searching revealed that one used an electro-mechanical machine [66] and the other only measured passive limb tone [67]; therefore, the latter review was also excluded.

One included review [41] recommended a single neurological evaluation (Hammersmith Infant Neurological Examination (HINE) [68]) as being appropriate to evaluate active and passive muscle tone in children aged 2 months to 2 years. The measurement properties were evaluated, but they required updating. We sought studies (not already evaluated in Goo and colleagues’ review [41]) that potentially included children with developmental central hypotonia (e.g., high-risk or pre-term infants) and provided additional reliability and/or validity evidence. Twenty articles were identified and eleven were included [69,70,71,72,73,74,75,76,77,78,79]. Studies including only typical infants [68,80] or only children with hypertonia [81,82] were excluded. Five other studies [83,84,85,86,87] did not add reliability or validity evidence. See Figure 1 for the flow diagram illustrating the search.

### 3.1. Included Reviews, Syntheses, and Individual Tools

Three systematic reviews met the inclusion criteria [15,41,42]. Two specifically aimed to identify characteristics of children with hypotonia and relevant measurement tools or techniques [15,42], while the other aimed to evaluate psychometric properties of muscle tone assessments [41]. Two related individual studies [14,22] explored measurement properties of select clinical assessment techniques; while one report [14] was included in a systematic review [42], the measurement properties were not evaluated. Another individual study explored associations between a different hypotonia assessment protocol and motor development in high-risk infants [23]. In addition, an ordinal scale defining muscle tone in children from severe hypotonia to severe hypertonia was identified in a textbook [45] and a ten-point hypotonia screening tool for children 1–5 years of age (MPH-10) was described in conference proceedings [46,47]. Finally, an evidence-based clinical algorithm for comprehensive assessment of children up to age 5 years [24] was identified, along with four studies supporting its development and appraisal [18,20,21,44], in addition to the starting point of Naidoo’s systematic review [15] and the study protocol [43].

One individual study [23] specifically evaluated children with central hypotonia and described the assessment criteria. The two related studies [14,22] described the development of a standardized protocol for the identification of hypotonia in children with a diagnosis where hypotonia may result from a combination of peripheral and central origins. However, the authors suggest that their protocol may be useful for children with other genetic disorders who likely meet the description of developmental central hypotonia. In a chapter related to assessment of children with CP, Howle [45] proposed criteria for grading active and passive muscle tone as mild, moderate, or severe. The MPH-10 [46,47] provides cut-off scores with recommendations for referral to specialist evaluation where appropriate. The preliminary six-point scale was published in conference proceedings [47], and the more developed ten-point scale (MPH-10) is attached in Appendix C.

Naidoo’s 2013 systematic review [15] did not distinguish between peripheral and central hypotonia in the search for evaluation methods, and this was also a limitation of the two survey/consensus studies that the author identified as providing the most relevant evaluation data [17,19]. Likewise, Govender/Naidoo and colleagues’ follow-up/algorithm development studies [18,20,21,44] and the clinical algorithm [24] include items relevant to distinguishing between central and peripheral hypotonia, as well as criteria to evaluate the level of functional impairment, regardless of the underlying cause.

De Santos Moreno and colleagues’ exploratory review [42] aimed to identify characteristics and assessment techniques relevant to children with central hypotonia, although many included articles focused on differentiating central and peripheral origins, and several focused on peripheral hypotonia, with only a brief description of central hypotonia in the introduction. The authors did not limit the search to peer-reviewed literature and provide a comprehensive map of clinical characteristics and assessment tests or techniques reported. Many characteristics/techniques were reported in descriptive or expert opinion articles or were secondary to the study purpose. No distinction was made between characteristics relevant to acutely ill neonates and those for older infants, and a number of the included articles were either updates of other included articles with almost identical content or data was taken from the introductory paragraphs where authors were merely citing some of the other 45 included articles.

Goo and colleagues’ psychometric properties review [41] included tools to evaluate both increased and decreased muscle tone and contrasts with the two other reviews as it focused only on standardized neurological and developmental assessments that included items or a subscale for the evaluation of muscle tone. A strength, relevant to this umbrella review, was that it separated measurement tools for neonates from those for older infants and young children. The only tool authors recommended for children 2 months to 2 years was the HINE, and they concluded that it is supported by at least moderate validity evidence. For children over 2 years of age, no tools were recommended as having adequate validity and being suitable to evaluate both active and passive muscle tone. See Table 1 for the included study details.

### 3.2. Quality and Risk-of-Bias

Authors of the three reviews and the mixed-methods algorithm study declared no conflicts of interest. One review was unfunded [42], while the other declared various non-commercial grants and scholarly funding sources [15,43,44]. One of the mixed-methods studies specifically stated that funding sources had no input into study conduct [20]. No review reported conflicts of interest or funding sources for their included studies. The three published articles [14,22,23] all reported no conflicts of interest, and one [22] indicated that the study was unfunded. The remaining tools were published in a textbook and conference proceedings. Howle [45] is a founder of Kaye Products Inc, a manufacturer of adaptive positioning and mobility devices, and wrote the CP chapter based on her PT expertise. One author of MPH-10 [46,47] has worked as an educational consultant for a manufacturer of standing and stepping devices. However, there was no commercial influence on MPH-10 development, which was unfunded.

The earliest systematic review by Naidoo [15] did not define the question or inclusion criteria adequately. The intent was to identify clinical tools and characteristics, but studies describing neuro-imaging and medical tests were also included. Using the ROBIS tool, this review was rated at high risk-of-bias in relation to study eligibility and unclear risk in relation to the identification and selection of studies. In other regards, this review was well conducted with duplicate reviews at all stages, use of the Oxford Centre for Evidence Based Medicine (OCEBM, 2009) evidence rating for diagnostic or differential diagnoses tests [90], and use of the Assessment of Methodological quality of Systematic reviews (AMSTAR) quality rating [91] to structure and self-evaluate strengths and weaknesses. It was rated at low risk-of-bias for data collection and study appraisal and for synthesis and findings. Since the findings were not overstated, and the author identified and accounted for the limitations in study identification in the analysis and follow-up article [18], this review was rated low risk-of-bias overall.

The exploratory review by De Santos Moreno and colleagues [42] had a similar broad question and search strategy, with duplicate review at all stages of the search, data extraction, and appraisal. However, only two articles related to the mixed-methods clinical algorithm study were included [18,20], without any justification of why the remaining studies (and the algorithm itself) were not, calling into question the strategy, sources, and criteria. The 60 excluded articles are not identified, and reasons for each exclusion are not provided. An appraisal of study conduct was completed using relevant guidelines for each study design. However, it appears that the level of evidence was defined using intervention design (OCEBM, 2011) [92], and it is unclear how or why Grading of Recommendations Assessment, Development and Evaluation (GRADE) [93] was used. There is no indication that the quality aspects of GRADE were considered, and the single included randomized controlled trial (comparing strength training versus regular PT on strength and balance) [94] was rated as providing higher level evidence than the other 44 studies, although review authors acknowledge that hypotonia was not evaluated. In addition, this review lacks clear recommendations or directions for future research. This review was rated as high risk-of-bias on each domain and overall.

The psychometric properties review by Goo and colleagues [41] had a clear question and inclusion criteria with duplicate search and screening. However, it is unclear whether data extraction and synthesis was conducted in duplicate, and a protocol was not published in advance. Data were summarized appropriately using visual charts to compare included tone items. Inclusion of relevant items and the COSMIN checklist [88] were used to substantiate the recommendation of the HINE for children 2 months to 2 years and the need to develop relevant tools to quantify muscle tone in older children. Using the ROBIS tool [32], risk-of-bias was unclear for the study eligibility criteria and low for remaining domains with low overall risk-of-bias.

The AGREE II tool [33] was used to rate the quality of Govender/Naidoo and Joubert’s clinical algorithm development. In comparison to the published report of the expert review [44], our consensus differed somewhat with some lower ratings, although ratings of scope, rigor of development, and editorial independence were almost identical. The greatest difference relates to the perception of applicability, likely due to the different purpose of the algorithm as a tool to diagnose children with hypotonia from any cause. For children with developmental central hypotonia, some aspects relevant to peripheral hypotonia are less relevant, and more specifics are required to evaluate or quantify hypotonia in order to identify potential developmental trajectory. In addition, we rated stakeholder involvement lower, since parent perceptions were not sought.

See Figure 2 for ROBIS and Figure 3 for AGREE II consensus results. MMAT was used to evaluate the conduct of individual studies, including those that were part of the mixed-methods clinical algorithm; the results are reported in Table 1. Full details of all quality ratings may be found in Appendix A.

### 3.3. Comparison of Studies, Characteristic,s and Tools across Reviews and Single Studies

Review purposes and inclusion criteria differed significantly. Goo and colleagues’ psychometric properties review [41] included only neurological or developmental tests and overlap of included studies with the other two reviews occurs only in the introduction. Naidoo’s review [15] excluded expert opinion evidence but combined these articles with the two most relevant studies identified [17,19] in the follow-up paper where the first section took the form of a scoping literature review [18]. De Santos Moreno and colleagues’ exploratory review [42] included 45 studies but concluded that only 4/45 attempted to measure or define hypotonia. Two [18,20] were part of the clinical algorithm mixed-methods study [24], and data from the other two [17,19] were a major part of the data used for algorithm development. However, the clinical algorithm [24], systematic review [15], and preliminary algorithm [21] were not identified among the 45 studies. See Table 2 for a comparison of included studies across reviews. In order to focus on studies relevant to this umbrella review question, articles on neonates were excluded, as well as articles that only briefly mentioned characteristics of hypotonia in their introductory paragraphs.

Table 3 compares characteristics and assessment methods reported in each included study or tool. Characteristics only listed in one review/study/tool or specific to peripheral hypotonia were excluded. Shaded rows indicate selection in six out of ten or more studies/tools.

Despite being widely acknowledged as being subjective, and dependent on examiner experience, decreased resistance to passive movement was the most commonly reported method overall. It was not included in the two tools (HINE and MPH-10) aiming to quantify muscle tone using objective methods. The HINE passive shoulder elevation item measures increased resistance to passive movement primarily, but a score of 1 (significant deviation from normal) can be given if there is “no resistance”, which may be interpreted as decreased resistance. Head lag on pull to sit and excessive hip abduction ROM or frog leg lying posture were next in frequency with each being reported in seven out of ten studies/tools. These were also the objective methods determined to correlate most strongly with the expert rating of hypotonia in one tool development study that provided moderate quality validity evidence [14].

De Santos Moreno and colleagues’ exploratory review [42] included every item included in Table 3, other than excessive shoulder flexion and head bob in sitting. Reflex testing was excluded as it is specific to peripheral hypotonia. Reflex testing is included in Govender/Naidoo’s clinical algorithm study [24] and the HINE for the purpose of differentiating between peripheral and central muscle tone disorders. Other objective assessment techniques reported in at least five studies/tools include excessive ROM in ankle dorsiflexion, slip through hands on vertical or axillary suspension, and assessment of motor skills using a valid and reliable standardized evaluative measure.

### 3.4. Measurement Properties of Included Tools and Measures

Howle’s proposal [45] appears to be the earliest to quantify hypotonia as being mild, moderate, or severe and considers both active and passive aspects of muscle tone. Content validity is supported by expert opinion but lacks specific and objective clinical methods resulting in a subjective overall rating that is heavily dependent on examiner experience. It was used in one study [106] by an author of the exploratory review [42] to help identify those children with Down syndrome most at risk for hip subluxation/dislocation. However, no formal validation studies were completed, and psychometric properties would be rated as poor using OMRF criteria.

The MPH-10 [46,47], provides cut-off scores distinguishing children having mild/moderate or severe hypotonia. It was developed based on a literature review and expert opinions and includes all the objective clinical methods of assessment reported five or more times in Table 3. It is also the only tool to suggest criteria for inclusion of results from a standardized motor skill assessment and to provide guidelines to quantify characteristics relevant for children over 2 years such as leaning on external supports and decreased activity tolerance. However, it lacks some aspects of postural assessment identified in the clinical algorithm mixed-methods study, and, since no formal reliability and validity studies have been completed, the MPH-10 would also be rated on OMRF as poor. Segal and colleagues [23] assessed children aged between 6 and 12 months and distinguished topography of hypotonia, with a score of 1 (present) for each region (neck, trunk, and limbs) for a maximum score of 3. This provides some limited quantification of severity, as children with hypotonia affecting all three regions presumably were more functionally impaired than children who only presented with hypotonia in one region. One assessor conducted all muscle tone assessments, and reliability was not evaluated. Content validity is not discussed; however, the tests for neck and trunk tone (pull to sit and ventral suspension) are among the most highly reported in the literature [42] and in Table 3. The techniques for assessing limb hypotonia (limb resistance, while opposing passive movement, and floppiness on limb shake-up), however, are only passive, and it is unclear how the examiner determined that infants of this age (mean 9 months) were actively providing resistance to passive movement. According to OMRF criteria, this scale may be considered to have poor reliability and validity.

Soucy and colleagues [14] identified objective measurement techniques with the best association with identification of hypotonia via an expert clinician in children aged 1–7 years. The most strongly associated items were head lag on pull to sit and hip abduction range of motion (ROM) > 60 degrees. These are also among the most highly reported characteristics, although it is often described as observation of frog leg posture, rather than objective measurement of ROM. This study was well-conducted and validity evidence would be described as sufficient but of moderate quality due to its smaller sample size. Other items associated with hypotonia included vertical suspension (slip through hands) and increased ROM in ankle dorsiflexion, that are also highly reported in Table 3. Inter-rater reliability was explored in the earlier study by Wessel and colleagues [22]. Although insufficient over the entire one-year study, reliability was sufficient after 6 months experience and supported by moderate quality evidence for the items: response to passive movement; vertical suspension (slip through hands); and pull to sit with head lag. COSMIN risk-of-bias and modified GRADE ratings for both studies are noted in Table 1. See Appendix D for a summary of validity and reliability results. Quality and risk-of-bias rating details may be found in the online Appendix A.

#### 3.4.1. Evidence-Based Clinical Algorithm

Naidoo/Govender and Joubert’s mixed-methods study was comprehensive and the resulting clinical algorithm [24] is structured around distinguishing peripheral and central causes of hypotonia. It also provides a quantification of functional impairment that is relevant for children with developmental central hypotonia and describes items to be considered in postural assessment such as W sitting, winging of scapulae, rounded shoulder posture, and protruding abdomen that are excluded or not emphasized in other tools. It also suggests using a functional assessment to quantify lack of endurance or activity tolerance. This algorithm is relevant for children 1–5 years, and may help to fill a gap, since Goo and colleagues’ [41] review did not identify any tools for children over 2 years that have adequate validity and address both active and passive aspects of muscle tone. AGREE II ratings were positive with concordance between the expert review rating published by the author [44] and our rating, considering differing purposes. COSMIN ratings are not appropriate for the algorithm since it is a guideline and measurement properties could only be evaluated for any sub-tools added to it.

#### 3.4.2. Hammersmith Infant Neurological Examination (HINE)

For children 2 months to 2 years of age, the HINE was recommended by Goo and colleagues [41] as having eight items evaluating aspects of active and passive muscle tone and demonstrating at least moderate validity as rated using the COSMIN 2010 checklist [88]. Since the publication of that review, the COSMIN tool has been updated to provide risk-of-bias ratings and guidance for clinician-administered outcome measures. A brief search (first 20 items) of Google Scholar and PubMed in November 2023 for the term ‘Hammersmith Infant Neurological Examination’ identified a number of newer studies reporting validity and reliability properties for the HINE.

Moderate quality evidence demonstrates sufficient (intraclass correlation coefficient (ICC) 0.79–1.00) inter- and intra-rater reliability in high-risk or pre-term infants [72,75,77,79]. Sufficient and high-quality evidence supports the predictive validity of global HINE scores in relation to later CP diagnosis [70,78], cognitive impairment [73,74], and motor delay [71]. Sufficient and high-quality construct validity evidence supports anticipated associations with magnetic resonance imaging (MRI) [79] and intelligence quotient (IQ) [76]. Sufficient, low-quality (due to sample size) evidence supports associations with motor, language, and cognitive developmental scales [77]. In addition, clinical feasibility and utility data have been reported [69]. It should be emphasized that, although these studies are likely to have included some children with developmental central hypotonia, psychometric properties of the HINE specifically for this population are not yet known. See Appendix D for COSMIN risk-of-bias and GRADE rating summaries. Scoring details may be found in online Appendix A.

## 4. Discussion

This umbrella review evaluated three systematic reviews and a mixed-methods study that synthesized the evidence and developed a clinical algorithm for assessment of children with hypotonia. One systematic review [15] was part of the mixed-methods study and contributed towards development of the algorithm [24]. One review focused on psychometric property evaluation of standardized developmental or neurological examinations with muscle tone sections [41]. It recommended only one tool (HINE) for evaluation of children aged 2 months to 2 years and concluded that valid, reliable tools for evaluation of children over 2 years are yet to be developed. The most recent review [42] was rated as high risk-of-bias but provided a comprehensive map of characteristics of children with hypotonia and assessment methods reported within descriptive and expert opinion literature and concluded that valid, reliable tools for quantification of hypotonia are needed.

In addition, five studies/tools were identified that had not been evaluated within these published reviews. Two studies provide moderate reliability evidence [22] for the use of subjective measures by experienced raters and moderate validity evidence [14] for the use of head lag (pull to sit) and measurement of hip abduction >60 degrees as objective criteria for the identification of hypotonia. These two criteria are also the most highly reported objective methods identified in this review. A third study [23] provides criteria to evaluate the topography (neck, trunk, and limbs) of hypotonia but not severity and has poor reliability and validity. The remaining tools are the only ones to attempt quantification of the severity of hypotonia, but one is based on subjective criteria [45] and neither have undergone formal validation processes and are rated poor for psychometric properties at this time. The MPH-10 [46,47] is suitable for children over two years and includes objective methods but requires further development.

Hypotonia is still not well defined or understood, and there are many aspects of disagreement within the literature and noted within included reviews and syntheses. The remainder of the discussion will be structured around discussion of these debated definitions, associations, and assessment methods. This will be followed by recommendations for clinical practice and future research.

*Decreased resistance to passive movement* continues to be the most cited definition of hypotonia and the most reported assessment method in this review. However, there are many difficulties beyond the subjective aspects of the assessment. These include difficulties in distinguishing decreased resistance due to low tone or decreased stiffness in the muscle, different visco-elastic properties of tendons/ligaments, decreased limb inertia, increased ability to relax muscles, increased ability to avoid involuntary muscle activation, and differing muscle activation patterns [58].

*Palpation* is an examination technique used to evaluate the resistance of underlying muscle tissue to the examiner’s finger and is less commonly cited than resistance to passive movement as a means of assessing muscle tone [108]. This term is used in the psychometric properties review [41] as one means of assessing resting or passive muscle tone, along with observation, and is distinguished from ROM measurement and assessment of resistance to passive movement. However, Naidoo/Govender’s mixed-methods studies list palpation and resistance to passive movement together. In the clinical algorithm [21,24], the assessment question or red flag is as follows: ‘Is there decreased resistance to passive movement?’ The preferred assessment test or technique, in brackets, is palpation. This confusion in terminology is also seen in the Delphi consensus study [20].

One study that did not meet the inclusion criteria [59] described a nine-point scale (Arms Legs Trunk (ALT) scale) using a combination of palpation and resistance to passive movement to assess passive muscle tone in children with Down syndrome. Tone is rated from -4 (severe hypotonia) to +4 (severe hypertonia), with 0 indicating normal tone, and includes descriptors for slight, mild, and moderate differences. Although inter-rater reliability (between scale developer and one trained rater) appears sufficient (k = 0.82–0.92), this was based on a very small sample of a maximum of six children. Although study authors suggest further research to develop this tool and evaluate measurement properties, no reports of this were identified.

Distinguishing *hypotonia with and without weakness* is a common theme in the numerous expert opinion-based articles focused on differentiating central and peripheral origins of hypotonia. In the exploratory review [42], authors note the confusion in the literature where strength is reported to be near normal in central hypotonia, while surveys report decreased strength as being one of the most highly reported characteristics of children with hypotonia [17,19]. This confusion may relate to the lack of distinguishing between children with central or peripheral hypotonia or lack of specifying that, while strength may be decreased in comparison to children who are typically developing, this differs from paralytic weakness and the lack of anti-gravity movement seen in children with peripheral hypotonia. One expert-opinion study describes the use of the scarf sign test in infants with hypotonia, noting whether the hips flex against gravity during this maneuver. Children with developmental central hypotonia such as Prader Willi syndrome may have excessive scapular movement indicating severe hypotonia, but will still demonstrated anti-gravity movement in the hips, while children with, e.g., spinal muscular atrophy remain in the frog leg resting position without movement [109].

*Diminished or absent deep tendon reflexes* are a strong indication of a peripheral or lower motor unit disorder [20]. Measurement is not directly relevant to the quantification of hypotonia in children with developmental central hypotonia, but assessment should be considered in children with severe hypotonia and weakness, along with referral for appropriate medical evaluation. Although reflex testing was recommended in Naidoo [15,18], Govender [24], and in the HINE, it may be considered an earlier step in the evaluation of children with developmental central hypotonia, as a means of excluding other diagnoses.

*Hypermobility versus hypotonia* is difficult to distinguish, and survey participants agreed that they are frequently seen together, although their exact relationship is unknown [17,19]. Excessive ROM in hip abduction and ankle dorsiflexion are objective means of evaluation but do not necessarily distinguish joint hypermobility, ligamentous laxity, and muscle extensibility. Hypermobility scales such as the Beighton or Bulbena scales are used for assessment of individuals with connective tissue disorders [110]. Although the Beighton score is more well-known, the Bulbena scale is more relevant for children with developmental central hypotonia, since it evaluates hypermobility in hips, knees, and ankles [111].

*W or M sitting* where the child sits on the floor with the bottom between the feet, knees in front, and feet out to the side is often used by children with developmental central hypotonia, as the wider base of support provides increases stability [112]. This position is also common when femoral anteversion peaks in typically developing children (3–6 years) and usually decreases or disappears at older ages [113]. W sitting is included in the postural assessment section of the mixed-methods clinical algorithm [24] and is also noted in the recent exploratory review [42]. Inclusion in the evaluation of children over 2 years is supported by the survey [18] and expert consensus [20], and may suggest joint hyperlaxity in combination with hypotonia [4].

However, the only reference for W sitting in Naidoo’s systematic review [15] is Carboni and colleagues [52] long-term, follow-up description of 41 individuals diagnosed with hypotonia with favorable outcome (formerly known as benign congenital hypotonia). In that study, authors specifically describe participants as easily able to W sit with their ‘bottom on the heels. In such a posture feet are passively forced to maximal plantar flexing’ [52] (p. 385). This is not W sitting, as typically understood, and suggests that the participants in that study may have more in common with connective tissue disorders than the typical population with developmental central hypotonia [99]. While young children with hypotonia are described as having rounded back posture [4,99], rounded shoulders [17,19], and leaning on supports [114], Carboni and colleagues’ participants were much older and able to walk and run without difficulties. This may be another reason why they displayed different characteristics, such as shortening in the lumbar muscles and triceps surae, as well as hyper-lordosis and winging of the scapulae.

The link between hypotonia and *delayed or decreased motor skills* is also debated in the literature [20]. One study included in this review [23] measured a significant link between hypotonia and delayed motor development, while an earlier study [61] found no connection. One difference is that Segal and colleagues [23] measured both active and passive aspects of muscle tone, while in Pilon and colleagues’ study [61] hypotonia/hypermobility was defined using passive ROM only. An early cohort study [115], reports delayed motor skills as being typical in children with congenital hypotonia. The MPH-10 suggests that a delay of 25% (2 standard deviations from the mean) in motor skills may be associated with mild/moderate hypotonia, and survey and consensus studies [17,18,19,20] including therapists and physicians also suggest that this link is commonly seen in clinical practice, although the exact nature of the relationship is still unknown.

### 4.1. Recommendations for Clinical Practice

Combined with the evidence supporting methods for quantification of hypotonia (Table 3), the validity and reliability evidence reported by Goo and colleagues [41], and that updated in this review, the HINE appears to be the most appropriate tool for the assessment of infants with suspected hypotonia aged 2–24 months. It is a simple, quantifiable, and widely studied measure [70] and, together with the Prechtl General Movement Assessment (GMA) and neuroimaging, it plays a crucial role in diagnosing and prognosing CP, as recommended by international clinical practice guidelines [116,117]. Use of the HINE has notably increased in recent years in some locations [118,119].

As seen in Table 3, in addition to the muscle tone section, the HINE also contains a number of other items that have been shown to be characteristics of children with hypotonia. These include facial appearance, facial response and oral motor control, head and trunk posture, active movement, kicking in vertical suspension, and whole-body response to lateral tilting. While infants and children with hypotonia may not qualify for early intervention using developmental milestones alone, HINE cut-off scores could assist in determining eligibility for an emerging CP diagnosis or being ‘at-risk’ for motor or intellectual disabilities. These infants are often dismissed and families told to “wait and see” while waiting years for a rare diagnosis to be identified. The co-diagnosis of CP, where relevant, could help families understand predicted gross motor trajectories and appropriate evidence-based interventions. 

For children over two years of age, tools for hypotonia evaluation continue to require development. Naidoo/Govender’s clinical algorithm, can be used to guide comprehensive assessment. The MPH-10 could be used in combination with the algorithm to provide quantification of hypotonia. Valid and reliable standardized motor assessments should be used to quantify motor delay and the potential developmental trajectory. Examples of possible age-normed assessments include the following: Developmental Assessment of Young Children 2nd Edition (DAYC-2) motor domain; Brigance Inventory of Early Development III motor domain; Peabody Developmental Motor Scale (PDMS-2); Alberta Infant Motor Scale (AIMS); Tests of Infant Motor Performance (TIMP); Bayley Scales of Infant and Toddler Development 4th Edition (BSID-IV) motor subscale, etc.

Children with 10–25% delay in motor development may fit the profile of mild/moderate hypotonia. Interventions recommended to promote motor skills in children with Down syndrome and other causes of developmental central hypotonia should preferably be child-active and incorporated into natural routines. Specific examples may include early tummy time, early kicking, treadmill training, and sensory-motor activities [51,57,120].

Children who are 50% delayed in motor skills, or with a HINE score <40 [70], are high-risk of functioning at Gross Motor Function Classification System (GMFCS) IV and V [121] and may fit the profile of severe hypotonia. Recommendations for children with CP, GMFCS IV, and V should be implemented, including early introduction of adaptive seating [122,123], supported standing [124], stepping [125], and power mobility devices [126,127,128], to promote ON-Time positioning and mobility experiences [129]. Consideration should also be given to monitoring hip health and inclusion in surveillance programs, since non-ambulant children with hypotonia are also at high risk for hip displacement [130,131].

### 4.2. Recommendations for Research

The HINE’s total score is derived from 26 items related to cranial nerve function, posture, movements, tone and reflexes, and reactions. Many items correspond directly with commonly reported findings in children with hypotonia, as illustrated in Table 3. The 2017 version of the HINE proforma [132] is not copyright and has been in the public domain for more than 5 years. It has been previously modified by adding an asymmetry sub-score to help identify children with hemiplegia whose function may not fall below cut-off scores relevant to children with bilateral CP [81].

Rather than develop a free-standing assessment or measurement tool for developmental central hypotonia, we propose an exploration of a HINE hypotonia sub-score, providing additional valuable insights for both clinical and research purposes. The HINE could be administered in full as per the standard protocol, and the hypotonia sub-score could then be recorded and interpreted as an additional score, in a similar manner to the asymmetry score.

A recently published study [133] (since our search for additional studies reporting psychometric properties of the HINE) suggests that the sub-scores ‘reflexes and reactions’, ‘cranial nerve function’, ‘movements’, and the asymmetry score combined may be more predictive of CP diagnosis than total HINE score. In contrast, the greatest number of items relevant to children with developmental central hypotonia are to be found in the muscle tone and posture sections. This supports our hypothesis that a different sub-score may be relevant and assist in the evaluation of severity and identification of potential functional trajectory in children with developmental central hypotonia.

See Appendix E for HINE items proposed as being relevant to the assessment of children with developmental central hypotonia, based on the results of this overview. Expert consensus studies would be beneficial to further validate the content of this proposal. A score of 2.5–3 is suggested as normal, 2.0–2.4 might indicate minimal hypotonia, 1.9–1.5 moderate and, <1.4 severe. These proposed ranges would need to be validated by a large cohort study, that would ideally be conducted prospectively. However, since the HINE is already used extensively around the world, centers that have the raw data from studies completed could potentially score the hypotonia sub-score retroactively, leading to a large amount of data on outcomes of children with developmental central hypotonia.

For children with developmental central hypotonia, over 2 years of age, presenting to OT and PT services, there are currently no recommended tools. The MPH-10 requires further development but could be combined with aspects from the clinical algorithm [24] and Howle’s scale [45]. A starting point for this could be expert consensus as part of face and content validation studies.

### 4.3. Review Limitations

The primary limitation of an umbrella review/overview is the risk-of-bias of the included reviews and the quality of synthesis within each review. Although one systematic review rated highly for risk-of-bias, it was essentially a scoping review and identified a large number of descriptive and expert opinion articles reporting characteristics and methods of assessment relevant to our review question. Since so few reviews were identified, and the lower risk-of-bias reviews were completed a few years ago, we chose to also search for individual studies that had not been included in the reviews to identify tools relevant to this population.

Although we did limit our search in one database to systematic reviews published since 2000, only narrative reviews appear to have been published earlier and these were already considered in our included reviews. We did not limit language in any database and did not limit publication type or date in several databases. We also included a review that covered Spanish language journals and another review that included references more than 5 decades old. However, it is possible that we missed tools or reviews published in other languages or in the grey literature.

Due to the heterogenous nature of our reviews and tool development studies, a variety of study quality and risk-of-bias tools were used. Also, the COSMIN methodology for a systematic review of measurement properties was not relevant to compare all tools, as so few formal tools were identified. As a result, limited comparisons across tools and studies were possible. Instead, we focused on a descriptive comparison of included studies, identifying components of measurement methods relevant to children with developmental central hypotonia and summarizing suggestions for clinical practice and future research.

## 5. Conclusions

This overview of the best available evidence identified three systematic reviews, a clinical algorithm, and five individual studies proposing tools for the evaluation of children with developmental central hypotonia. The HINE is recommended for the evaluation of children 2 months to 2 years and items for a potential hypotonia sub-score are proposed. Measurement properties of the HINE, specifically for children with developmental central hypotonia need to be established, and validation studies for the content and potential cut-off scores for the hypotonia sub-score are needed. Further research is needed to combine individual tools and methods identified in this review with aspects of the clinical algorithm to create a new tool for children over two years of age with developmental central hypotonia.

## Figures and Tables

**Figure 1 healthcare-12-00493-f001:**
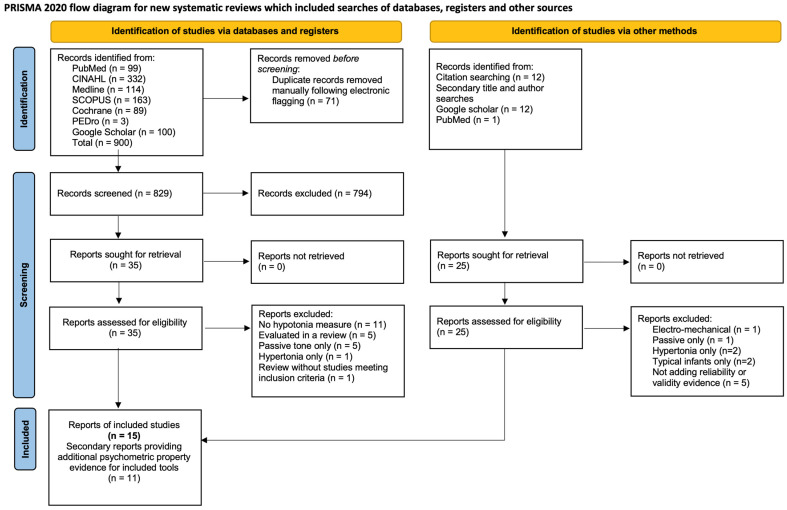
Prisma 2020 flow diagram of the search process.

**Figure 2 healthcare-12-00493-f002:**
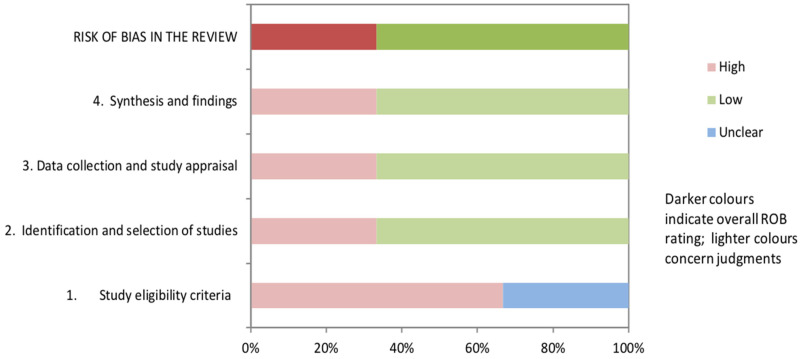
ROBIS risk-of-bias ratings for the three systematic reviews [15,41,42].

**Figure 3 healthcare-12-00493-f003:**
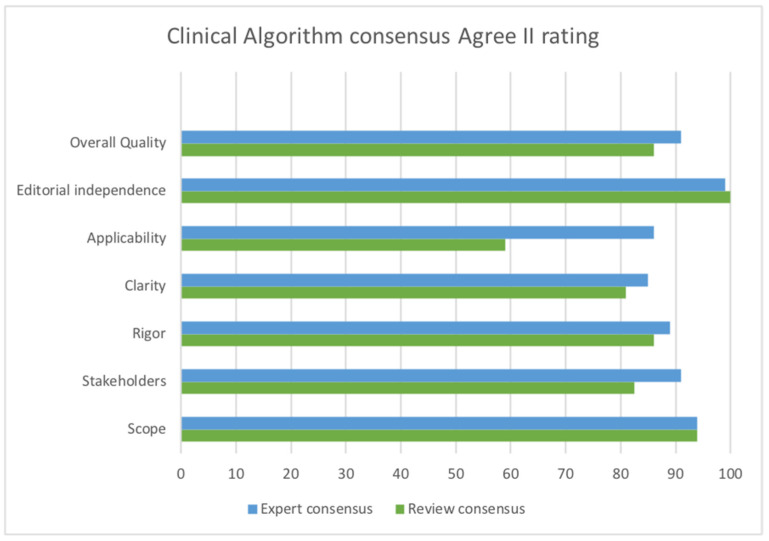
AGREE II rating comparing consensus of published expert review [44] and consensus of this review.

**Table 1 healthcare-12-00493-t001:** Included studies.

CitationCountry	Study Design	Purpose	Details	Results	Conclusions
Quality/ROBEvidence Level
**Systematic or scoping reviews**		
**Naidoo 2013a**[15]**South Africa****ROBIS****Low risk** 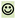	Systematic Review	Identify and appraise existing assessments for children reported in the literature. Identify gaps to inform future research.	*Search:* Medline, CINAHL, ERIC, ScienceDirect, GoogleScholar, and PEDro—database inception to January 2013*Terms:* (defin* OR assess* OR test OR evaluat*) and (hypotonia OR low muscle tone) and (children)*Inclusion:* Peer-reviewed studies (OCEBM evidence levels for diagnostic studies 1–4 only) including children 0–12 years with low muscle tone. Excluded 15 expert opinion references.	Twelve studies included. Only two [17,19] investigated criteria or characteristics useful to evaluate hypotonic status. Assessment components identified: history, observation, neurologic examination, decreased strength, decreased activity tolerance, delayed motor skills, rounded shoulder posture, leaning on supports, increased flexibility, hypermobile joints, and poor attention/motivation.	Limited evidence available regarding the most reliable or valid methods or tests for the assessment of hypotonia in children. Need for further research.
**Goo et al., 2018**[41]**Australia****ROBIS****Low risk** 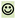	Systematic review	Identify and examine psychometric properties of muscle tone assessments for children 0–12 years.	*Search:* PubMed, Medline, CINAHL, and Embase—January 2000—November 2017*Terms:* ‘muscle’ AND ‘tone’ OR ‘tonus’ OR ‘tonic’ OR ‘stiff*’; OR ‘neurologic*’ AND ‘motor’ OR‘neuromotor’ OR ‘neurosensory’ OR ‘neurodevelopmental’ OR ‘neurobehavior’; AND ‘assess*’ OR ‘evaluat*’ OR ‘measur*’ OR ‘test’ OR ‘tests’ OR ‘testing’ OR ‘examin*’; AND ‘child*’ OR ‘infant*’ OR ‘neonat*’; AND ‘psychometric’ OR ‘reliab*’ OR ‘reproducib*’ OR ‘valid*’ OR ‘agreement’.*Inclusion:* Quantitative clinical assessment of resting and/or active tone in children 0–12 years with manual and psychometric properties available.	A total of 21 assessments reported over 97 articles. Divided assessments according to three age groups.Ten measures for neonates.Six measures for infants 2 months—2 years.Five measures for children >2 years.COSMIN 2010 checklist [88] evaluated reliability and validity of all measures.Extracted and compared items and techniques rating active and passive muscle tone across tools.Recommended only tools having at least moderate validity and/or reliability that measured both active and passive muscle tone.	HINE was the only tool recommended for children aged 2 months to 2 years. Measures both active and passive tone (8 items) and has at least moderate validity (*content validity and reliability not reported from pre-2000 studies*).New measures required for children >2 years. NSDMA has only one passive and one active tone item and limited validity.
**De Santos-Moreno et al., 2020**[42]**Spain****ROBIS****High risk** 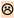	Exploratory systematic review	To describe characteristics associated with hypotonia and identify methods used for diagnosis in children	*Search:* PubMed/Medline, PEDro, Cochrane, BVS, IBECS, MEDES, Web of Knowledge, and ScienceDirect. Dates of search unclear.*Terms:* “quantitative evaluation”, “muscle tone”, “symptom assessment”, “floppy muscle”, “muscle hypotonia”, “infant”, and “child.” Combined using AND/OR.*Inclusion:* Articles describing characteristics of children with hypotonia or tests of hypotonia assessment—excluding those exclusively for peripheral hypotonia.	Forty-five studies included: twenty-eight expert opinions, narratives, or literature reviews, one RCT, three case-control, two case studies, and eleven observational studies.GRADE level of evidence appraised as low or very low in 44/45 studies.A total of 4/45 studies presented clinical maneuvers with no evidence of validity or reliability—[5,9,14,89]. Only 4/45 studies attempted to measure hypotonia or define characteristics [17,18,19,20].	No valid and reliable test or scale identified for the diagnosis and quantification of hypotonia.Most common methods: observation; pull to sit; frog leg posture; vertical suspension; ventral suspension; and scarf sign.Relationship of muscle strength, hypermobility, and maintenance of anti-gravity postures with hypotonia are debated in the literature.
**Synthesis—and associated studies**			
**Govender and Joubert 2018**[24]**South Africa****MMAT****5/5**	Mixed-methods—published over six studies [15,18,20,21,24,44];Protocol [43]	Develop and validate a clinical decision-making process to guide clinicians during the assessment of children with hypotonia from genetic or neurologic origins.	Systematic Review [15]Survey of 319 clinicians [18]Delphi consensus: 11 experts [20]Qualitative critique: 59 clinicians [21]Expert critique: 11 experts [44]	Evidence-based algorithm includes history, clinical assessment (posture, dysmorphic features and drooling, anti-gravity tests, resistance to passive movement, motor skills, muscle strength, endurance and activity tolerance, and reflex testing), and activity and participation limitation evaluation. BSF, activity, and participation impairments are rated from 0 (no impairment) to complete (present 95% time and disrupting daily life)	Supports comprehensive assessment of hypotonia from all causes. Aspects still to be developed:specifics of each assessment and quantification of ‘degree of hypotonia’; clinical utility evaluation and implementation resources;adaptation for culture and setting; and monitoring and auditing aspects.
**Naidoo 2013b**[18]**South Africa****MMAT****5/5**	Survey of current practices based on descriptive analysis of tests and characteristics (reported in this publication) and systematic review [15]	To determine the current practices of OT, PT, and Pediatrics in the assessment of hypotonia. Survey content developed from literature review (17 studies, level 1–5 evidence).Eleven characteristics and nine assessment methods used as a survey tool basis. Survey developed with four additional experts and piloted with five clinicians.	Eleven characteristics: increased flexibility; decreased resistance to passive movement; delayed motor skills; leaning on external supports; decreased activity tolerance; W or M sitting postures; difficulty in prone or supine postures; winging of scapula; diminished or absent reflexes; hypermobile joints; and frog-like postures.Nine assessment methods: observation; palpation; resistance testing; posture; manual muscle testing; reflex testing; developmental tests; antigravity tests; and range of movement. Survey responses received from 319 OTs, PTs, and Pediatricians.	Methods of assessment used and characteristics reported (% responses): (1) Observation—96%.(2) Postural assessment—91%: leaning on supports (76.3%); W or M sitting (75.9%); scapulae winging (73.8%); rag doll postures (70.9%); difficulties with antigravity postures (72.5%).(3) Palpation—73%.(4) Decreased resistance to passive movement—63.4%.(5) Increased flexibility—57.2% (6) Hypermobile joints—68.8%.(7) Reflex testing—32%:decreased/absent reflexes (30.3%).	Identified need for more objective measures in assessments of hypotonia.A total of 78% respondents identified the need to quantify hypotonia.Ages 2–5 described as a population challenging to assess accurately.
**Naidoo and Joubert 2013**[20] **South Africa****MMAT****5/5**	Delphi consensus study	Generate consensus on the assessment of hypotonia in respect to the clinical characteristics, tests, and methods identified from the literature and survey study	Eleven opinion leaders and clinical researchers in OT, PT, and Pediatricians working with children in South Africa.Two-round Delphi consensus	Consensus above 70% achieved on 11 aspects of the assessment: reflexes; palpation; motor skills; muscle strength; balance; range of motion; endurance; anti-gravity positions; postural assessment; drooling/oral-motor; and myopathic facies.	Palpation, assessment of posture, and anti-gravity positions ranked most highly.Diminished/absent reflexes and difficulties with anti-gravity positions were the most sensitive signs of peripheral origins of hypotonia.
**Govender and Joubert 2016**[21]**South Africa****MMAT****4/5**	Qualitative, emergent– systematic focus group design	Evaluate and critique a draft clinical algorithm based on results of phase-1 studies [15,18,20]	Fifty-nine clinicians (OT, PT, and Pediatricians). Ten focus groups from various locations in South Africa.	Evidence-based, holistic assessment that can be used across ages. Clarifications and algorithm layout changes suggested.	Need for further clarification of terms and quantification of severity.
**Govender 2018**[44]**South Africa****MMAT**NA	Expert critique	Systematically appraise the clinical algorithm prior to clinical implementation	Ten clinical and academic experts (clinicians, policy makers, and guideline developers) in OT, PT, and Pediatrics based in South Africa	A total of 9/10 recommended adoption without modification.Overall assessment—91%Scope and purpose—94%Stakeholder involvement—91%Rigor of development—89%Clarity of presentation—85%Applicability—86%Editorial independence—99%	Further development: (1) Measurement criteria and definitions, required for implementation.(2) Detail on resource implications, monitoring, and auditing.(3) Intervention resources based on assessment results.
**Individual measurement tools—not evaluated in reviews or syntheses**		
**Howle, 1999**[45]**USA****OMRF****Poor***Reliability**Validity*(no data available)	Descriptive/Expert; Opinion*Decision making in Pediatric**Neurologic Physical Therapy* textbookCerebral Palsy chapter p 23–37;Tool p37	Describes seven-point ordinal scale for muscle tone evaluation: −3—severe hypotonia;−2—moderate hypotonia;−1—mild hypotonia;0—normal tone;+1—mild; hypertonia+2—moderate hypertonia; and+3—severe hypertonia.	Score −3, severe hypotoniaActive: inability to resist gravity; lack of co-contraction of proximal joints for stability; and apparent weakness. Passive: no resistance to movement imposed by the examiner; full or excessive passive ROM; and hyperextensibility.Score −2, moderate hypotoniaActive: decreased tone primarily in axial muscles and proximal muscles of the extremities and interferes with length of time posture can be sustained. Passive: very little resistance to movement when imposed by the examiner; less resistance encountered in movement around the proximal joints; and hyperextensibility at knees and ankles upon weight-bearing.Score −1, mild hypotoniaActive: interferes with axial muscle co-contractions; delays initiation of movement against gravity; and reduces speed of adjustment to postural change. Passive: some resistance to joint changes; full passive ROM; and hyperextensibility limited to joints of hand, ankles, and feet.
**Morgan and Paleg 2012**[46,47]**USA****OMRF****Poor***Reliability**Validity*(no data available)	Tool development	Introduce Morgan Paleg Hypotonia Scale (MPH-10)	Ten items each scored on a three-point ordinal scale:Score 0—typical function;Score 1—mild/moderate impairment;Score 2—severe impairment.Divide raw score by total number of items tested to obtain overall score.	1. Vertical suspension;2. Prone (ventral) suspension;3. Head lag (pull to sit),4. Hip abduction;5. Ankle dorsiflexion;6. Scarf sign;7. Shoulder posture;8. Leaning onto supports;9. Activity tolerance;10. Motor abilities.	<0.5 no significant hypotonia.**Suspect**: 0.5–1.2 reflects mild/moderate hypotonia; referral to a pediatric PT/OT recommended.**Fail**: 1.2–2 reflects severe hypotonia; recommend referral to qualified specialist, e.g., developmental pediatrician.
**Wessel et al., 2013**[22]**USA****COSMIN****ROB**Adequate*Inter-rater reliability***GRADE****±**Moderate	Tool development;Inter-rater reliability	Determine the reliability of hypotonia diagnosis (as a potential clinical indicator of glioma in children with NF1).	Fifty-six children 1–7 years with NF1 were assessed by Ped/Nurse (non-therapist) and PT for presence of hypotonia using vertical suspension and resistance to passive movement at knee and elbow. In addition, PT used pull to sit and recorded presence or absence of head lag.	*Inter-rater reliability* on subjective hypotonia assessment was insufficient overall:76% (37/49); k = 0.485. However, reliability improved over the year:first 6 months 63% (15/24); sufficient in last 6 months 88% (22/25) k = 0.746.	Hypotonia indicated by vertical suspension (slip through hands), response to passive movement, and pull to sit (head lag). Can achieve sufficient inter-rater reliability on these items with training.
**Soucy et al., 2015**[14]**USA****COSMIN****ROB**Very good*Criterion validity***GRADE****+**Moderate	Tool development;Criterion validity	Identify diagnostic criteria for assessing hypotonia in children with NF1.	Fifty-five children aged 1–7 years with NF1. Subjective assessment: vertical suspension (slip through hands); resistance to passive movement at elbow and knee; and muscle palpation—soft, normal, or rigid.Objective assessment: ROM—hip abduction, knee extension, and ankle dorsiflexion with knee extended; pull to sit (head lag).(triceps fat %; grip strength—hand-held dynamometer—>2 years only).	All measures except triceps fat % and knee ROM correlated with subjective impression of hypotonia.*Criterion validity*Presence of head lag paired with hip abduction >60 degrees resulted in highest correlation with expert therapist rating:sensitivity 80%;specificity 83%.	Head lag on pull to sit and hip abduction >60 degrees are clinically measurable and objective findings in hypotonic children with NF1. Research required to determine if these items are applicable to other pediatric populations in which hypotonia is a prominent clinical feature.
**Segal et al., 2016**[23]**Israel****MMAT****4/5****OMRF****Poor***Reliability**Validity*(no data available)	Analytical; Cross-sectional study	Describe relationship between central hypotonia and motor development. Determine the relative contribution of nuchal, truncal, and appendicular hypotonia domains to motor development.	A total of 164 children assessed by PT (PDI) and neurologist (tone) on the same day. A total of 128 children with central hypotonia. Thirty-six children with normal tone.Mean age 9.6 months ±4 months.Nuchal hypotonia: head lag on pull to sit, head bobbing or drop on sitting position, or head drop on ventral suspension (active). Truncal hypotonia: slipping on axillary suspension and back curving on sitting position (active).Appendicular hypotonia: decreased resistance while opposing passive movements; increased floppiness on limb shake-up (passive).	A total of 115 children had truncal hypotonia. Seventy had nuchal hypotonia.Ninety-three had appendicular hypotonia. Score 3—hypotonia in all three regions;Score 2—hypotonia in two regions;Score 1—hypotonia in one region.Central hypotonia was associated with motor delay. No correlation between muscle tone and later CP diagnosis (9/128 children).	Neck and trunk (axial hypotonia) most common.Motor delay strongly associated with neck (ß −0.6), and either trunk, and/or limb hypotonia (ß −0.4, *p* < 0.001).

BSF: body structure and function; CP: cerebral palsy; COSMIN: COnsensus on Standards of Measurement properties in health care INstruments; COSMIN rating: risk-of-bias rating for reliability or validity evidence; GRADE: COSMIN-modified GRADE rating of level of evidence for psychometric property evaluation; HINE: Hammersmith Infant Neurological Examination; MMAT; Mixed Methods Appraisal Tool; NF1: Neurofibromatosis type 1; OCEBM: Oxford Centre for Evidence Based Medicine levels of evidence; OMRF: McMaster Outcome Measures Rating Form; OT: occupational therapist; PT: physical therapist; Pediatrics/Pediatrician: Pediatrician or Pediatric Neurologist; PDI: Psychomotor Developmental Index (from Bayley II assessment); ROB: risk-of-bias; ROBIS: Risk-of-Bias In Systematic reviews; ROM: range of motion; USA: United States of America.

**Table 2 healthcare-12-00493-t002:** Comparison of studies identified in each review.

Study by First Author, Date, and [Citation]	Naidoo 2013a/b [15,18]	Goo et al., 2017 [41]	de Santos Moreno et al., 2023 [42]	Umbrella Review
Curran 1998 [95]				
Howle 1999 [45]				
Pilon 2000 [61]				
Carboni 2002 [52]				
Sender 2003 [96]	Duplicate of Gowda 2007			
VanToorn 2004 [97]				
Leyenaar 2005 [98]				
Martin 2005 [17]				
Gowda 2007 [99]				
Jan 2007 [100]				
Martin 2007 [19]				
Bodensteiner [5]				
Harris 2008 [6]				
Peredo 2009 [4]				
Jain 2011 [101]	Duplicate of Gowda 2007			
Paleg 2012 [46]				
Naidoo 2013a [15]				
Naidoo 2013b [18]				
Naidoo et al., 2013 [20]				
Wessel 2013 [22]				
Naidoo 2014 [43]				
Paleg 2014 [47]				
Hartley 2015 [102]				
Soucy 2015 [14]				
Bay 2016 [103]				
Christiansen 2016 [104]				
Govender 2016 [21]				
Kaur 2016 [105]				
Segal 2016 [23]				
de Santos Moreno 2017 [106]				
Govender 2018 [24]				
Kaler 2020 [107]				
Madhok 2022 [89]				

Studies are listed in date order from earliest to most recent. Thick black lines and grey shading indicate search end dates for each review. Light green shading indicates that the article briefly describes clinical assessment maneuvers such as head lag with pull to sit or ventral suspension. Dark green shading indicates that the listed article reports a research study related to hypotonia assessment or measurement tools. In the umbrella review column (this review), studies not already fully evaluated in the other reviews are noted.

**Table 3 healthcare-12-00493-t003:** Characteristics or methods identified in each review, study, or tool.

Citation/Tool		Type	Method	Naidoo[15,18]	Goo[41]	DeSantos[42]	Govender[24]	Howle[45]	MPH-10[46,47]	Wessel [22]	Soucy [14]	Segal[23]	HINE
Age range				Children	0–12 yrs	Children	1–5 yrs	Children	1–5 yrs	1–7 yrs	1–7 yrs	6–12 m	2–24 m
HeadFaceOral-motor	Excessive drooling	P	O										
Facial tone	P	O										
Alertness/ facial response to visual/auditory/social stimuli	A	O										
Head bob/drop in sitting	A	O										
Pull to sit or head lag	A	Clinical										
Axial/Trunk	Kyphosis, rounded shoulders Decreased trunk extension	PA	OO										
Weakness	A/P	O/MM										
Shoulder, arm and hand	Scarf/shoulder adduction	P	Clinical										
Shoulder elevation/flexion	P	Clinical										
Slip thru hands	A/P	Clinical										
Winging of Scapulae	A	O										
Hip and leg	↑ hip abduction ROM‘Frog leg’ resting posture	AP	ROMO										
‘W’ or ‘M’ sitting/posture	A	O										
Popliteal angle ROM	P	ROM										
↑ knee extension ROM	P	ROM										
↑ Ankle Dorsiflexion ROM	P	ROM										
Whole Body	↑ Flexibility/hypermobility	P	ROM										
↓ Proximal co-contraction	A	O										
Whole body anti-gravity responses	Vertical suspension	A	Clinical										
Lateral Tilting	A	Clinical										
Ventral suspension/‘Rag Doll’	A	Clinical										
Overall supine/prone	A	Clinical										
Decreased endurance	Decreased activity tolerance	A	O										
Leaning on external supports	A	O										
Decreased/slow movements	A	O										
Decreased resistance to passive movements	P	Clinical										
Deep tendon reflex testing	P	Reflex										
Muscle palpation	P	Palp										
Delayed motor skills	A	ST										

*Type of tone being tested:* A: active; P: passive; ↓: decreased; ↑: increased; yrs: years; m: months. *Method of Testing:* Clinical: assessing response to clinical assessment maneuvers; MM: manual muscle testing; O: observation; Palp: palpation; ROM: range of motion; ST: standardized valid and reliable motor assessment—except for HINE (includes a record of motor skills, but this section is not scored).

## Data Availability

All data are available in the paper or appendices.

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
