# Peer review of "Identifying and Evaluating Young Children with Developmental Central Hypotonia: An Overview of Systematic Reviews and Tools"

_healthcare, 2024, doi:10.3390/healthcare12040493_

Round 1

Reviewer 1 Report

Comments and Suggestions for Authors

The topic of the review is of great interest and relevance in the clinical field. However, some changes need to be made to the manuscript to improve it.

In the Introduction section the previous background that led to the completion of the work must be described. The information given in paragraph 2 on page 2 is methodology. The objectives of the study are very extensive. I recommend that these be more concrete and clear.

Regarding the Results, how many articles were included? The number of articles included in the Flow diagram does not correspond to the number of articles shown in Table 1.

Regarding Table 1, is it necessary? The table is very extensive and mixes different studies. Could it be included as supplementary material?

Table 2 is difficult to understand. Include table footer with explanation and remove this from the text.

It is recommended to simplify the text and synthesize the information. The text is often difficult for the reader. This makes you forget what the objective of the work is.

Author Response

Thank-you for your review. Please see attachment for a response to each of your comments.

Reviewer 2 Report

Comments and Suggestions for Authors

1. Abstract is incomplete without the result and conclusion part, please include both the parts in the abstract.

2. Kindly explain the pathophysiology and possible etiology of hypotonia in the introduction part

Author Response

Thank you for your review. Please see attachment for responses to your comments.

Reviewer 3 Report

Comments and Suggestions for Authors

It has been a pleasure to review this review on assessment tools for central hypotonia.

The only aspect that we must point out as an improvement is the review registration link, which does not work, it must be reviewed. I have searched for it manually by the first author, but I can't find it.

The way in which the results of the Quality and risk-of-bias section are expressed seems very accurate to me.

The differentiation that the authors make regarding the implications of these results for clinical practice or research is very important.

Author Response

Thank you for your review. Please see attachment for our response to your comments.

Round 2

Reviewer 1 Report

Comments and Suggestions for Authors

The authors have done improvements in the article based on comments from the first review. 

I consider the manuscript to be suitable for publication.